# Gender-transformative health promotion interventions for linking and retaining tuberculosis-diagnosed adult men in care in sub-Saharan Africa: A scoping review protocol

Siyabonga Kave[1]*, Yandisa Sikweyiya[1,2,3], Nelisiwe Khuzwayo[1]

1 School of Nursing and Public Health, College of Health Sciences, Howard College Campus, University of KwaZulu-Natal, Durban, South Africa, 2 Gender and Health Research Unit, South African Medical Research Council, Pretoria, South Africa, 3 School of Public Health, University of the Witwatersrand, Johannesburg, South Africa

* siyakave@gmail.com

## Abstract

Tuberculosis (TB) remains a leading cause of infectious disease mortality globally, with men in sub-Saharan Africa (SSA) disproportionately affected. Men experience higher TB incidence, delayed care-seeking, poorer treatment adherence, and elevated mortality compared to women. Gender-transformative health promotion interventions that challenge harmful masculine norms, enhance health system responsiveness, and foster positive health behaviours show promise for improving linkage to and retention in TB care. However, evidence on their design, implementation, and effectiveness in SSA is limited. Following the Arksey and O'Malley framework, refined by Levac et al., this scoping review will map evidence on gender-transformative health promotion interventions targeting adult men (≥18 years) diagnosed with any form of TB in SSA. We will search PubMed via MEDLINE, the Cochrane Library, Embase.com, Global Health, Web of Science, PsycINFO, Google Scholar, Africa-Wide Information, and the WHO Library, and screen reference lists of included studies. Eligibility will follow the Population–Concept–Context (PCC) framework, focusing on interventions supporting linkage to care, treatment initiation, adherence, or retention in care. Data will be charted using a standardized extraction tool, quality appraised using the Mixed Methods Appraisal Tool, and narratively synthesized using the RE-AIM (Reach, Effectiveness, Adoption, Implementation, Maintenance) framework. Stakeholder consultations will inform the interpretation of findings. As a protocol, no results are reported. The review will identify intervention characteristics, implementation strategies, barriers, facilitators, and outcomes across SSA contexts. Findings will guide the development of culturally sensitive, gender-responsive TB interventions aligned with the Sustainable Development Goal 3 to improve linkage and retention among men diagnosed with TB in SSA.

which permits unrestricted use, distribution, and reproduction in any medium, provided the original author and source are credited.

**Data availability statement:** No datasets were generated or analysed during the current study. All relevant data from this study will be made available upon study completion.

**Funding:** The author(s) received no specific funding for this work.

**Competing interests:** I have read the journal's policy and the authors of this manuscript have no competing interest.

## Background

Tuberculosis (TB) remains one of the world's deadliest infectious diseases, causing an estimated 1.25 million deaths annually and ranking just below Human Immunodeficiency Virus/Acquired Immunodeficiency syndrome (HIV/AIDS) in global mortality [1,2]. Despite advances in diagnosis and treatment, TB disproportionately affects low- and middle-income countries, particularly in sub-Saharan Africa (SSA) [3,4].

Globally, TB incidence and mortality are consistently higher among men than women [5,6]. In 2021, six million men developed TB compared to 3.4 million women, with men accounting for 54% of TB deaths among HIV-negative individuals in the African Region [7,8]. In South Africa, pulmonary TB prevalence among men aged 15 years and older was 1.6 times higher than in women [3,9], while a multi-country study across 56 nations reported a male-to-female TB incidence ratio of 1.8 [10].

Evidence from SSA further indicates that men are less likely than women to complete TB treatment and more likely to experience treatment default or failure [4]. For example, men had a 1.3–1.8 higher risk of treatment interruption in multi-country SSA studies [2], and in South Africa, men accounted for 60% of TB treatment interruptions with lower overall treatment completion rates compared to women [11]. Consequently, men face higher risks of TB complications and mortality despite the effectiveness of early diagnosis and treatment strategies [12,13].

Gender disparities in TB outcomes arise from an interplay of biological, psychosocial, behavioural, and health system factors [14]. Biological sex differences, including genetic and hormonal influences, affect immune responses and TB susceptibility [15]. Psychosocial determinants such as dominant masculine norms, imprisonment, homelessness, alcohol and substance use, smoking, and HIV co-infection, further elevate men's risk [7,16].

Dominant masculine norms where entrenched ideals of stoicism and self-reliance hinder timely diagnosis, treatment initiation, and adherence [3,16] may discourage men from seeking care or expressing vulnerability, contributing to delayed diagnosis, poor treatment initiation, and high loss-to-follow-up rates [11,8]. System-level barriers, including feminized healthcare environments, unwelcoming staff attitudes, and clinic hours incompatible with work schedules [17,18] also impede men's engagement with TB services.

Nevertheless, global frameworks emphasize the need to implement gender-specific TB programmes [1,19]. When combined with health promotion interventions that reduce stigma, increase access, and sustain engagement in care, these strategies enhance men's participation across the TB care continuum [16]. In addition, co-developing interventions with men further improves cultural relevance, acceptability, and sustainability [6].

While progress in TB diagnostics and surveillance has advanced, equivalent investments in gender-sensitive programming remain insufficient, particularly in contexts where hegemonic masculinity norms impede care-seeking [20]. For instance, in SSA, TB interventions often remain gender-blind or gender-neutral, overlooking how masculinity shapes care-seeking behaviours [3,10].

These disparities underscore the need for interventions that improve linkage, adherence, and retention specifically among men [1,21]. Indeed, emerging evidence suggests that gender-transformative interventions designed to challenge harmful masculine norms, foster supportive health systems, and promote positive health behaviours offer promise in improving TB outcomes among men [4,18]. However, most evidence comes from high-income settings, and there is limited synthesized knowledge on the design, implementation, and outcomes of preventive and curative gender-transformative health promotion interventions for men with TB in SSA, including interventions that are sensitive to contextual and cultural factors and engage key actors involved in care delivery [22,6].

Given these challenges, mapping the design, implementation, and outcomes of gender-transformative health promotion interventions in SSA is essential to inform culturally and contextually appropriate strategies that enhance care engagement and treatment success among men [3,7]. Along this line, a scoping review is warranted to map and synthesize existing evidence on the design, implementation, and outcomes of gender-transformative health promotion interventions aimed at improving linkage to and retention in care among adult men diagnosed with all forms of TB in SSA. This protocol article reports on the methods that will be used when conducting that scoping review.

## Methods

This scoping review forms part of a PhD study titled 'Developing a Gender-Transformative Health Promotion Intervention for Men Diagnosed with TB in the Eastern Cape Province, South Africa. Ethical approval was not required for this scoping review, as it involves the synthesis of data from publicly available literature and does not include human participants. However, the overarching PhD study has been approved by the Biomedical Research Ethics Committee (BREC) of the University of KwaZulu-Natal (BREC Ref No: 00008572/2025) (S1 Fig).

Reporting of the scoping review will adhere to the Preferred Reporting Items for Systematic Reviews and Meta-Analyses extension for Scoping Reviews (PRISMA-ScR) [23] to ensure methodological transparency and completeness (S2 Fig).

The scoping review's methods are recorded according to the framework for scoping reviews proposed by Arksey and O'Malley [24] and refined by Levac et al. [25]. This framework includes six steps: 1) identifying the research question, 2) identifying relevant studies, 3) study selection, 4) charting the data, 5) collating, summarizing, and reporting results, and 6) consultation with stakeholders on study findings.

### Identifying the research question

A preliminary literature review was conducted to inform the development of a focused research question for the proposed scoping review. The central research question guiding this review is: What evidence exists on gender-transformative health promotion interventions that aim to strengthen linkage to and retention in care among adult men diagnosed with TB in SSA?

To address this overarching question, the review seeks to explore the following sub-questions:

1. What are the barriers and facilitators, including cultural and contextual factors, influencing gender-transformative health promotion interventions?

2. What methods, strategies, and delivery platforms are used to implement these interventions?

3. What are the outcomes of these interventions on linkage, treatment initiation, adherence, and retention in care?

### Identifying relevant studies

Relevant literature on men diagnosed with TB will be searched through electronic databases, including PubMed via MEDLINE, Cochrane Library, Embase.com, Global Health, Web of Science, PsycINFO, Google Scholar, Africa-Wide Information, and the WHO library database [26].

These databases were strategically selected for their relevance to the scope of this review, which focuses on gender-transformative health promotion interventions for men diagnosed with TB in SSA. MEDLINE, Embase, Global Health, and the Cochrane Library were included due to their comprehensive coverage of biomedical and public health research, ensuring that clinical, epidemiological, and intervention studies relevant to TB management and health promotion were captured.

WHO Regional Libraries and Africa Wide Information were specifically selected to enhance geographic representation and provide access to region-specific studies and grey literature from high TB burden countries in SSA. PsycINFO was included for its extensive coverage of psychological, behavioural, and mental health research, which is essential for understanding gendered behaviours, adherence patterns, and psychosocial components of health promotion interventions [1,6].

In addition to peer-reviewed publications, grey literature including government reports, policy briefs, and program evaluations will be included. These sources often provide valuable information on real-world implementation, contextual factors, and outcomes of TB interventions that may not be captured in peer-reviewed literature. Inclusion of grey literature ensures a more comprehensive mapping of gender-transformative interventions [5].

To ensure comprehensive coverage, the literature search will include studies published from 2000 to 2024. This extended timeframe allows the review to capture earlier interventions that may inform contemporary gender-transformative TB programs and identify changes in intervention design, implementation, and outcomes over time.

The preliminary search strategy (S3 Fig) was developed in consultation with a trained medical librarian (LR) to ensure thorough literature coverage. It will focus on combinations of keywords and controlled vocabulary related to key concepts including "tuberculosis," "men" or "male populations," "gender-transformative interventions," "health promotion," and "sub-Saharan Africa" [15], with the latter now expanded to include the full list of countries classified within the region according to the World Bank [27]. The primary search will be conducted in PubMed (MEDLINE), and the complete preliminary PubMed search string (Table 1) is presented below to enhance transparency and replicability.

## Study selection

Study eligibility will be assessed by two reviewers (SK and YS). They will first screen titles and abstracts of all records identified through the search. Full-text articles deemed potentially relevant will then be retrieved and assessed independently by the same reviewers using the predefined inclusion and exclusion criteria. Any disagreements during either stage will be resolved through discussion. If consensus cannot be reached, a third reviewer (NK) will serve as an adjudicator. To guide this process, we will apply the Population, Concept, and Context (PCC) framework recommended by Pollock et al. [28] and aligned with JBI guidelines [23]. The use of that framework will ensure that the eligibility criteria capture studies involving the relevant populations (adult men diagnosed with TB), focus on gender-transformative health promotion interventions, and are situated within appropriate contexts to answer the research questions. Detailed inclusion and exclusion criteria are presented in Table 2 below:

## Charting the data

Data from all included studies will be systematically extracted using a standardized charting tool developed by the review team (S4 Fig). Key information to be extracted includes study type, location, and purpose; participants' sociodemographic characteristics, such as age, cisgender male status, health status, and socioeconomic status (including self-rated health and health-seeking behaviours) [29]; study design and setting, including where TB was diagnosed and treated (community versus hospital, and whether the care was provided at a specialized TB centre); and the type of care providers in charge of TB (specialist versus generalist clinicians). Additional data will include the presence or absence of anti-discrimination and TB gender-specific treatment policies, as well as statistics from relevant inspection services on the effective implementation of these policies. TB-related details will cover the type of TB, treatment scheme and duration, and whether

**Table 1. Databases, rationale, and associated search string.**

| Database | Rationale | # | Search strategy |
|---|---|---|---|
| Pubmed (MEDLINE) | Broad coverage of biomedical literature and comprehensive indexing with MeSH. Essential for TB clinical, epidemiological and intervention studies and for retrieving MeSH tagged literature on Africa (e.g., "Africa South of the Sahara"). | #1 | ("Tuberculosis"[MeSH] OR "Mycobacterium tuberculosis"[MeSH] OR tuberculos*[tiab] OR TB[tiab] OR XMTB[tiab]) |
| | | #2 | ("Men's Health"[MeSH] OR "Masculinity"[MeSH] OR men[tiab] OR male[tiab] OR masculinity[tiab] OR "gender norms"[tiab] OR "gender roles"[tiab]) |
| | | #3 | ("Health Promotion"[MeSH] OR "Health Behavior"[MeSH] OR "Behavioral Research"[MeSH] OR "Health Education"[MeSH] OR "Gender Equity"[MeSH] OR "health equity"[tiab] OR "gender-transformative"[tiab] OR "gender transformative"[tiab] OR "gender-transforming"[tiab] OR "gender-transformative interventions"[tiab] OR "health promotion interventions"[tiab] OR "behavioral interventions"[tiab] OR "behavioural interventions"[tiab]) |
| | | #4 | ("Scoping Review"[tiab] OR "scoping review protocol"[tiab] OR "Stop TB"[tiab] OR "TB elimination"[tiab]) |
| | | #5 | ("Africa South of the Sahara"[MeSH] OR "Sub-Saharan Africa"[tiab] OR "Sub Saharan Africa"[tiab] OR SSA[tiab] OR "Angola" OR "Benin" OR "Botswana" OR "Burkina Faso" OR "Burundi" OR "Cabo Verde" OR "Cameroon" OR "Central African Republic" OR "Chad" OR "Comoros" OR "Congo, Democratic Republic" OR "Congo, Republic" OR "Côte d'Ivoire" OR "Equatorial Guinea" OR "Eritrea" OR "Eswatini" OR "Ethiopia" OR "Gabon" OR "Gambia" OR "Ghana" OR "Guinea" OR "Guinea-Bissau" OR "Kenya" OR "Lesotho" OR "Liberia" OR "Madagascar" OR "Malawi" OR "Mali" OR "Mauritania" OR "Mauritius" OR "Mozambique" OR "Namibia" OR "Niger" OR "Nigeria" OR "Rwanda" OR "Sao Tome and Principe" OR "Senegal" OR "Seychelles" OR "Sierra Leone" OR "Somalia" OR "South Africa" OR "South Sudan" OR "Sudan" OR "Tanzania" OR "Togo" OR "Uganda" OR "Zambia" OR "Zimbabwe") |
| | | #6 | #1 AND (#2 OR #3 OR #4) AND #5 |

The principal investigator (SK), together with the primary supervisor, (YS), and the co-supervisor, (NK) will ensure the validity of the search strategy to be used.

complementary or alternative medicines were used. Economic information will include direct and indirect costs of TB care and the source of funding (government subsidies versus other payers, including out-of-pocket payments).

Data on treatment adherence and reasons for stopping TB treatment (e.g., death, loss to follow-up, treatment failure, adverse drug reactions, or patient withdrawal) will also be extracted. Health promotion interventions will be examined, including barriers and facilitators to implementation, type of intervention (preventive versus curative; gender-specific or not; culturally sensitive or not; informed by local contextual issues), duration of follow-up, and reasons for stopping follow-up. Finally, participants' outcomes will be recorded, including survival versus more complex outcomes such as quality of life, employment opportunities, and role conservation or modification within participants' communities. A pilot test will be conducted on a sample of studies to refine the extraction categories and ensure clarity and consistency. The principal investigator (SK) will conduct the primary data extraction. The primary supervisor (YS) will independently verify the accuracy of all extracted data. The co-supervisor (NK) will serve as an independent reviewer to resolve discrepancies between SK and YS and will support ongoing refinement of the data-charting tool. The three authors will meet regularly throughout the extraction phase to review progress, resolve uncertainties, and ensure that the charting tool effectively captures all data relevant to the overarching research question.

The risk of bias of individual studies will be evaluated using the Mixed Methods Appraisal Tool (MMAT; S5 Fig) published in 2018 [30] to ensure data integrity, methodological rigor [31,28], and quality, alignment with study aims, and appropriate implications for future research.

## Collating, summarising, and reporting results

Data will be summarised narratively, consistent with the aims of a scoping review. Study characteristics will be presented using tables and figures, stratified by key variables including age, TB status, geographic location, and study design. We

**Table 2. Scoping review eligibility criteria based on the PCC Framework.**

| Domain | Inclusion | Exclusion |
|---|---|---|
| Population | Adult cisgender men (≥18 years) diagnosed with any form of TB, including:<br>• pulmonary,<br>• extra-pulmonary,<br>• latent, active,<br>• drug-sensitive, or<br>• drug-resistant TB.<br>Men across all stages of the TB care continuum, including:<br>• Newly diagnosed, on treatment, lost to follow-up, or re-engaged in care.<br>• No restrictions by socioeconomic status, race, or ethnicity. | • Studies involving populations other than adult men (≥18 years) diagnosed with TB.<br>• Studies focusing exclusively on women, children, or mixed populations without disaggregated data for adult men.<br>• Studies not reporting on linkage to, adherence with, or retention in TB care.<br>• Animal studies, laboratory-based studies, or biomedical research without human participants.<br>• Commentaries, editorials, opinion pieces, or conference abstracts without primary data.<br>• Studies focused solely on TB preventive therapy without linkage to treatment, care, or retention outcomes.<br>• Duplicate publications.<br>Non-English publications (if language restriction applies). |
| Concept | • Studies describing or evaluating gender-transformative health promotion interventions that support linkage to care, treatment initiation, adherence, or retention among adult men with TB.<br>• Interventions may be preventive or curative, reflecting all stages of the TB care continuum (2000–2024).<br>Eligible interventions include:<br>• Individual-level approaches (e.g., counselling, patient navigation, digital adherence technologies, peer support).<br>• Community-based or mass interventions (e.g., outreach, health education, media campaigns, group-based programs).<br>• Health system or implementation strategies integrating gender-transformative or behavioural components.<br>• Sources may include peer-reviewed studies, government documents, policy briefs, systematic reviews, or meta-analyses. | • Studies not focused on gender-transformative, health promotion, or behavioral interventions targeting adult men with TB.<br>• Interventions limited to biomedical or pharmacological treatment without a behavioral or adherence-focused component.<br>• Studies on general health system strengthening without relevance to linkage, adherence, or retention in TB care among men.<br>• Studies on preventive therapy only, with no linkage to TB treatment or care engagement.<br>• Commentaries, editorials, or abstracts without empirical or detailed intervention data.<br>• Studies lacking clear information on intervention strategies, delivery, influencing factors, or outcomes. |
| Context | • Studies conducted in healthcare or community settings related to TB care in sub-Saharan Africa (SSA), including TB clinics, hospitals, primary care facilities, or community health programs.<br>• Countries classified as low- or middle-income by the World Bank [27], at the time of the study (see Annexure S5 Table 3 for the list of eligible countries).<br>• Studies published between 2000 and 2024, with no language restriction (unless applied). | • Studies conducted outside sub-Saharan Africa.<br>• Studies published before 2000 or after 2024.<br>• Studies not specifying healthcare or community settings relevant to TB care.<br>• Basic science or laboratory research without clinical or public health relevance.<br>• Opinion pieces, editorials, or conference abstracts without primary or secondary data.<br>• Studies lacking essential contextual information (e.g., urban/rural setting) where required for analysis. |

will organize findings using Glasgow's RE-AIM framework [31] (Reach, Effectiveness, Adoption, Implementation, Maintenance). This will allow us to assess study reach, methodological diversity, diagnostic and intervention approaches, adoption into practice and policy, implementation of barriers and facilitators, and sustainability within health systems. Gaps in the literature will be identified through comparative analysis across study settings, intervention types, and outcomes to identify patterns and under-researched areas. This approach will inform future research priorities and the development of a gender-transformative intervention framework [15].

**Consultation with stakeholders on study findings.** The consultation phase will engage healthcare workers, TB experts, stakeholders, and men with lived experience of TB to ensure findings are contextually relevant and actionable [15] and can guide future locally tailored strategies.

**Definition of terms.** To ensure methodological rigor and consistency, the following key terms are defined based on recent evidence

- **Tuberculosis**: "a bacterial infectious disease caused by *Mycobacterium tuberculosis*, primarily affecting the lungs but capable of affecting other organs" [20].

- **Gender**: "refers to the roles, behaviours, activities, and attributes that a given society at a given time considers appropriate for men and women" [4].

- **Health:** "a state of complete physical, mental and social well-being and not merely the absence of disease or infirmity" [20].

- **Health promotion:** "is the process of enabling people to increase control over, and to improve, their health" [18].

- **Linkage to care:** "refers to the process of engaging a person with a disease to appropriate prevention, treatment, care and support services" [21].

- **Retention in care:** "refers to continuous, regular engagement of a patient, from the time of diagnosis, in an on-going comprehensive package of follow-up assessment, prevention, treatment, care and support services" [21].

- **Gender transformative interventions:** are approaches that seek to challenge gender inequality by transforming harmful gender norms, roles and relations through programmatic inclusion of strategies to foster progressive changes in power relationships between women and men [18].

**Data sharing.** In line with open science principles, the final data extraction sheet and metadata will be made publicly available via the Open Science Framework [16].

**Timeline.** The initial search will be conducted on January 31, 2026. Title and abstract screening, along with full-text screening, will be carried out from March to May 2026. Data extraction will be completed by July 2026, followed by analysis and synthesis of results from August to September 2026. As of October 2026, we anticipate finalizing the results and submitting the manuscript for publication by November 2026.

**Protocol deviation.** Any deviations from the outlined protocol will be fully documented and transparently reported in future publications and presentations of the study findings. The review team will ensure that any deviations are minimized, appropriately justified, and do not compromise the review integrity or intended objectives.

## Discussion

This scoping review will map and synthesize existing literature on gender-transformative and health promotion interventions in SSA designed to improve linkage to care and retention among men aged 18 years and older diagnosed with TB [2]. The review aligns with the World Health Assembly's 2014 resolutions and is embedded within the SDGs [2].

The proposed methods offer several anticipated strengths. Their structured and iterative approach is expected to provide a comprehensive overview of current interventions while maintaining transparency in study selection and data extraction [15,16]. Drawing on diverse data sources, engaging stakeholders, and employing robust quality appraisal tools will enhance the depth and contextual relevance of the findings [30]. Moreover, the use of a well-established synthesis framework, such as RE-AIM [24,31], will support the generation of practical insights to inform intervention design and policy development.

Nonetheless, certain methodological limitations are expected. Restricting the search to published and English-language studies may introduce selection bias and omit relevant evidence [22,15]. The heterogeneity of intervention designs and outcomes is also likely to limit comparability and preclude qualitative synthesis [17]. As with most scoping reviews, the focus will be on mapping rather than critically evaluating intervention effectiveness, which may constrain the strength of resulting policy recommendations [17].

To address these limitations, mitigation strategies such as iterative team discussions, piloting of data extraction tools, and triangulation with stakeholder perspectives will be employed, thus improve the reliability and applicability of the review findings [19,31,26].

## Dissemination plan

The findings of this scoping review will be submitted to PLOS ONE for publication. Additionally, the results will be shared through institutional websites and professional networks such as LinkedIn. Where feasible, preliminary findings will be presented at relevant conferences and meetings to engage the broader research and public health community.

## Conclusion

This scoping review will advance understanding of how gender-transformative and health promotion interventions have been designed, implemented, and evaluated to improve linkage to care and retention among men diagnosed with TB in SSA. By systematically mapping existing evidence, assessing intervention outcomes, and highlighting key knowledge gaps, the review will generate insights to guide the development of context-specific, gender-responsive strategies and inform future research, policy, and programmatic efforts aligned with the SDGs.

## Supporting information

**S1 Fig. Approval+letter+BREC+Ref + No + 00008572 + 2025 (1).**
(PDF)

**S2 Fig. PRISMA FLOW checklist.**
(DOCX)

**S3 Fig. Preliminary library search string by database.**
(DOCX)

**S4 Fig. DATA CHARTING form.**
(DOCX)

**S5 Fig. MIXED METHOD APPRAISAL tool.**
(DOCX)

**S6 Table. 3 Countries (LMIC) by the World Bank.**
(DOCX)

## Acknowledgments

Our gratitude goes to the School of Nursing and Public Health, Discipline of Public Health Medicine, University of KwaZulu-Natal for guidance and support in developing this scoping review protocol.

## Author contributions

**Conceptualization:** Siyabonga Kave, Yandisa Sikweyiya, Nelisiwe Khuzwayo.

**Project administration:** Siyabonga Kave.

**Supervision:** Yandisa Sikweyiya, Nelisiwe Khuzwayo.

**Writing – original draft:** Siyabonga Kave.

**Writing – review & editing:** Siyabonga Kave, Yandisa Sikweyiya, Nelisiwe Khuzwayo.

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
