## [Decision Letter · Decision Letter 0]

1 Sep 2025

Dear Dr. Kave,

Thank you for submitting your manuscript to PLOS ONE. After careful consideration, we feel that it has merit but does not fully meet PLOS ONE’s publication criteria as it currently stands. Therefore, we invite you to submit a revised version of the manuscript that addresses the points raised during the review process.

We look forward to receiving your revised manuscript.

Kind regards,

Mickael Essouma, M. D.

Academic Editor

PLOS ONE

Journal Requirements: 

2. Please amend either the title on the online submission form (via Edit Submission) or the title in the manuscript so that they are identical.

Additional Editor Comments:

The main limitation with this scoping review protocol is that the aim of the scoping review for which the protocol is being conceived is unclear. Along the same line, the authors did not clearly state in the Materials and Methods section whether the review will assess contextual and culturally-sensitive gender transformative health promotion interventions as well as the role of key actors involved in the conception and implementation of those interventions, although they mentioned the potential driving role of culture-specific factors for the predominance of tuberculosis in males and the global health community as an important audience for this research. See Lancet Glob Health 2025;13: e1627–35. Furthermore, it is unclear which countries they were referring to when reporting on LMICs in the manuscript. See BMJ Glob Health. 2022 Jun;7(6):e009704. doi: 10.1136/bmjgh-2022-009704 and https://doi.org/10.1007/s10741-025-10550-x.

It is important to specify throughout the manuscript whether the tuberculosis health promotion interventions you would like to assess are preventive and/or curative health promotion interventions, and whether you are addressing individual-level and/or mass/community-based interventions. Are you studying tuberculosis at large or a specific form of tuberculosis? This information should also be provided throughout the manuscript, including in Definitions sub-section will of the Materials and Methods section. See https://doi.org/10.1371/journal.pone.0322753 and Lancet Glob Health. 2024 May;12(5):e737-e739.

What is the relevance of the MMAT questionnaire in the supplemental material accompanying this submission? The data extraction chart provided as a suplemental material seems to be that of a critical interpretive synthesis whereas this protocol is for a scoping review. Pay attention to this matter.

Consider describing the Arksey and O'Malley methods for scoping reviews as did these authors (https://doi.org/10.1371/journal.pone.0314914) and integrating the comments made about review methods. Are you planning to add a «Consultation with stakeholders» sub-section in the Materials and Methods section as optional last step of the Arksey and O'Malley methods for scoping reviews?

Finally, consider conforming to PLOS One guidelines for formatting manuscripts. For instance, the abstract and references are not recorded as recommended by PLOS One both in the main text and in the reference section and numbering the manuscript's lines to ease its assessment. See how the abstract should appear in the manuscript: https://doi.org/10.1371/journal.pone.0314914.

More details about my comments are available from the attached document PONE-D-25-36889_reviewed by Mickael Essouma.pdf.

Mickael Essouma, M.D.

Reviewers' comments:

Reviewer's Responses to Questions

**Comments to the Author**

1. Does the manuscript provide a valid rationale for the proposed study, with clearly identified and justified research questions?

Reviewer #1: No

Reviewer #2: No

Reviewer #3: Yes

Reviewer #4: Yes

2. Is the protocol technically sound and planned in a manner that will lead to a meaningful outcome and allow testing the stated hypotheses?

Reviewer #1: Partly

Reviewer #2: No

Reviewer #3: Yes

Reviewer #4: Partly

3. Is the methodology feasible and described in sufficient detail to allow the work to be replicable?

Reviewer #1: No

Reviewer #2: No

Reviewer #3: Yes

Reviewer #4: Yes

4. Have the authors described where all data underlying the findings will be made available when the study is complete?

Reviewer #1: Yes

Reviewer #2: Yes

Reviewer #3: Yes

Reviewer #4: No

5. Is the manuscript presented in an intelligible fashion and written in standard English?

Reviewer #1: No

Reviewer #2: Yes

Reviewer #3: Yes

Reviewer #4: Yes

You may also provide optional suggestions and comments to authors that they might find helpful in planning their study.

Reviewer #1: The work entitled Gender-transformative and health promotion interventions for linking and retaining tuberculosis diagnosed adult men (18 years and above) in care in sub Saharan Africa: A scoping review protocol by Kave et al., intends to map and synthesize evidence on gender transformative and health promotion interventions that may support linkage to and retention in TB care among adult men (age 18 and above) in Sub Saharan Africa (SSA). Only in 2021, the African region reported 365.000 TB deaths among HIV negative people, with men accounting for 54% of these fatalities. Despite expanded access to TB services, men are less likely to engage in care due to structural, socio- cultural, and health system- related barriers. Gender norms associated with masculinity, such as stoicism, self reliance, and reluctance to seek help, negatively affect men's health- seeking behavior and retention in care. Please find my comments below.

It is well known that TB Globally affects men more than women, hence is not only an SSA condition. The added value of this review over what is published is missing. Until now, there are no significant biological experimental evidence supporting this gender increased risk. However, certain social conditions like imprisonment history or being homeless was not addressed here nor other comorbidities like being HIV positive, lung cancer, previous TB treatment, etc. Risk associated addictions like alcohol and drug abuse are missing also in this work. I believe that the increased risk of having TB and being male was not properly addressed in this work.

Reviewer #2: The study is about to provide a review protocol. Protocols are according to standard guidelines but to validate the protocol, study needs to be done in real-time. As authors have mentioned , the final study manuscript will be submitted by January 2026, that time it can be considered.

Authors can also look into statistical analysis of results obtained in meta-analysis for significance of interventions.

Reviewer #3: The protocol is well written and all the necessary details are provided. When arrried out, will provide importnat inputs to TB program mangers to address the gender speciifc access to high quality TB Care.

Reviewer #4: Authors are addressing a relevant subject to current End TB Strategy

My comments

BACKGROUND

- Could you provide evidence in numbers that men in SSA (or elsewhere, if no data available) were less likely to complete TB treatment, and more likely to default/fail treatment. I've come across a couple of articles from SSA and worth mentioning to strengthen the rationalisation of the study that aimed to look for interventions to improve retention in care.

METHODS

- Not sure you have provided the rationale for choosing only the past 10 years for your study. Have authors observed improvement in interventions over this period? You might have missed good interventions before this period and weaken/narrow your findings. I would probably extend the the period of search and skip the grey materials, as not necessarily peer-reviewed, unless if you rationalise this.

- Though mentioned in the limitations, confining your articles to those written in English might cause missing articles written in other languages in SSA that are crucial to your assessment.

- In the PRISMA flow chart, articles were deemed ineligible through automation tools- what are these tools?

- In your supplement, you have included SCOPUS database, but not mentioned in the text.

- In the supplement, the keywords and MeSH combination doesn't fit the aim of your study. In fact, it looks as if they belong to a different study! Nothing about the interventions you are after! This will lead to missing articles that you are looking for and the inclusion of heaps of articles that are not relevant to your study, wasting your time in reviewing them.

- Not sure why HIV is crucial for the study's database structure!

- The term sub-Saharan Africa might not explicitly be mentioned in each article's keywords and MeSH. I would consider searching each country separately to gain proper insight of what was done in each country.

**Do you want your identity to be public for this peer review?** For information about this choice, including consent withdrawal, please see our Privacy Policy

Reviewer #1: No

Reviewer #2: No

Reviewer #3: No

Reviewer #4: No

---

## [Author Response · Author response to Decision Letter 1]

13 Oct 2025

Dear Mickael Essouma and PLOS ONE team

We would like to sincerely thank you and the reviewers for the thoughtful and constructive feedback on our manuscript entitled manuscript title Gender-transformative health promotion interventions for linking and retaining tuberculosis-diagnosed adult men (18 years and above) in care in sub-Saharan Africa: A scoping review protocol, (Manuscript ID: PONE-D-25-36889). We greatly appreciate the time and effort invested in reviewing our work, and we believe that the comments have significantly strengthened the manuscript.

We have carefully considered all suggestions and have revised the manuscript accordingly. In the sections below, we provide a point-by-point response to each reviewer comment. All changes in the manuscript are clearly indicated in tracked changes, and we have highlighted the revisions where appropriate.

We hope that the revisions satisfactorily address the reviewers’ concerns and that the manuscript is now suitable for publication in PLOS ONE.

Sincerely,

Siyabonga Kave

---

## [Editor Report · Decision Letter 1]

4 Nov 2025

Dear Dr. Kave,

Thank you for submitting your manuscript to PLOS ONE. After careful consideration, we feel that it has merit but does not fully meet PLOS ONE’s publication criteria as it currently stands. Therefore, we invite you to submit a revised version of the manuscript that addresses the points raised during the review process.

We look forward to receiving your revised manuscript.

Kind regards,

Mickael Essouma, M. D.

Academic Editor

PLOS ONE

Journal Requirements:

Additional Editor Comments:

I. General comments

I have proposed a deletion of the statement (18 years and above) in the manuscript’s title to keep it shorter.

I have have proposed a reorganization of the introduction to improve its flow. I also proposed edits for the Methods and Discussion sections.

To avoid redundancy in the manuscript, I have proposed a deletion of the statement «To ensure a comprehensive coverage of the published literature, the review will include studies published from 2000 to 2024, allowing the capture of earlier interventions that may inform contemporary programs and assess changes in intervention design, implementation, and outcomes over time. This review will also explore contextual, cultural, and structural factors influencing intervention uptake and the role of key actors involved in intervention design and delivery.» in the introduction, leaving it only in the Methods section.

The text states that you included search terms related to sub-Saharan Africa in the search strategy. However, there are no terms related to sub-Saharan Africa in any table on search strategies provided in the supplemental material. Consider addressing that issue.

You seem to have implemented the eligibility criteria proposed in the first round of peer review. However, it is difficult to read your eligibility criteria, and there is inconsistency with the eligibility criteria currently listed, at least for the type of publications that will be excluded. As suggested in the first round of peer review, consider going through this article (https://doi.org/10.1371/journal.pone.0322753) and build your table on eligibility criteria exactly as they did to improve its readability.

Consider also rearranging tables’ numbers in the manuscript and in the supplementary material. Notably, consider inserting all tables (including those from the supplemental material) in a Word format rather than in a TIF format as seen in the current manuscript.

Consider updating references in the text and formatting references in the text and the reference section as observed in published PLOS One articles. Furthermore, there is some degree of citation gaming in this manuscript, and this issue should be addressed as well. Along this line, you keep mentioning the WHO End TB strategy throughout the manuscript whereas the period of evaluation of that strategy (2015 to 2020 is clearly over (see Lancet Infect Dis 2024; 24: 698–725), and reference 46 they used to back up their claims on the WHO End TB strategy is missing at the end of the manuscript. Therefore, consider removing all comments on the WHO End TB strategy in this manuscript.

II. Specific comments

They are appended to this decision letter under the name PONE-D-25-36889_R1_reviewed by Mickael Essouma.docx.

Mickael Essouma, M.D.
---

## [Author Response · Author response to Decision Letter 2]

1 Dec 2025

We sincerely appreciate the thoughtful and constructive comments provided by the editor and reviewers. A detailed, point-by-point response to all feedback is included in the accompanying rebuttal letter, where we indicate precisely how and where each comment has been addressed in the revised manuscript and supplementary materials.

---

## [Editor Report · Decision Letter 2]

4 Dec 2025

Dear Dr. Kave,

We look forward to receiving your revised manuscript.

Kind regards,

Mickael Essouma, M. D.

Academic Editor

PLOS ONE

Journal Requirements:

Additional Editor Comments:

In the hindsight, I came to realise that the WHO end TB strategy is ongoing, although ref 46 in the last version of the manuscript was misleading. See Https://www.who.int/teams/global-programme-on-tuberculosis-and-lung-health/the-end-tb-strategy.

Careful English language editing is required, removing typos and other language mistakes. For instance, what is the abbreviation of aub-Saharan Africa in this manuscript SSA or sub-SSA?

Conform to PLOS ONE referencing system.

The search strategy should include the complete list of SSA countries based on the World Bank classification. See for example systematic reviews on other conditions in SSA.

The eligibility criteria should be specified in the main manuscript. Therefore, move the table on eligibility criteria to the main manuscript.

In the data analysis sub-section, you need to learn state that data will be summarised narratively.

Mixkael Essouma, M.D.

---

## [Author Response · Author response to Decision Letter 3]

9 Dec 2025

Dear Dr. Mickael Essouma and the PLOS ONE Editorial Team,

We are pleased to resubmit our revised manuscript entitled “Gender-transformative health promotion interventions for linking and retaining tuberculosis-diagnosed adult men in care in sub-Saharan Africa: A scoping review protocol” (Manuscript ID: PONE-D-25-36889) for further consideration in PLOS ONE.

We would like to sincerely thank the editor for the valuable suggestions provided to improve the quality of our manuscript.

The following documents are included in this resubmission:

• The fully revised manuscript.

• A tracked-changes version highlighting all modifications.

• A detailed response letter addressing each comment.

• Updated figures, tables, and supplementary material.

Major revisions include:

• We acknowledge the important point regarding the WHO End TB Strategy. The earlier version of the manuscript contained phrasing that may have been misleading due to the way reference 46 was cited. We confirm that we have removed this ambiguity and clarified that the End TB Strategy is an ongoing initiative.

• We have also carefully revised the manuscript for English language accuracy, correcting typographical errors and ensuring consistent use of terminology. In particular, we standardised the abbreviation for sub-Saharan Africa to “SSA” throughout the manuscript, removing inconsistent forms such as “sub-SSA.”

• In response to the guidance on referencing, we have updated the entire manuscript to conform to the PLOS ONE referencing system.

• As recommended, we have incorporated the full list of Sub-Saharan African countries in the search strategy, based on the World Bank classification. This strengthens the transparency and reproducibility of the review process.

• We also appreciate the suggestion to specify the eligibility criteria within the main manuscript. The table on eligibility criteria has now been moved from the supplementary material into the main text.

• Finally, we have revised the Data Analysis subsection to clearly state that the data will be summarised narratively, with tables and figures used to present study characteristics and key findings.

• We are grateful for the editor’s constructive feedback, which has meaningfully improved the clarity and coherence of the manuscript.

We believe the revised manuscript now meets the PLOS ONE criteria and presents a clearer, more rigorous, and more methodologically sound scoping review protocol. We are confident that the revised version offers a strong and meaningful contribution to the literature on gender-transformative health promotion interventions for tuberculosis care in sub-Saharan Africa.

Thank you again for the opportunity to revise and resubmit our work. We look forward to your feedback.

Sincerely,

Siyabonga Kave

---

## [Editor Report · Decision Letter 3]

10 Dec 2025

Gender-transformative health promotion interventions for linking and retaining tuberculosis-diagnosed adult men in care in sub-Saharan Africa: A scoping review protocol

PONE-D-25-36889R3

Dear Dr. Kave,

We’re pleased to inform you that your manuscript has been judged scientifically suitable for publication and will be formally accepted for publication once it meets all outstanding technical requirements. Congratulations!

Kind regards,

Mickael Essouma, M. D.

Academic Editor

PLOS One

Additional Editor Comments (optional):

Consider providing the title of Table 2 in the main manuscript. Consider conforming to PLOS One referencing system; see how other authors cited references in their article: https://doi.org/10.1371/journal.pone.0322753.
---

## [Editor Report · Acceptance letter]

PONE-D-25-36889R3

PLOS One

Dear Dr. Kave,

I'm pleased to inform you that your manuscript has been deemed suitable for publication in PLOS One. Congratulations! Your manuscript is now being handed over to our production team.

Kind regards,

on behalf of

Dr. Mickael Essouma

Academic Editor

PLOS One